# Family Characteristics, Transgender Identity and Emotional Symptoms in Adolescence: A Population Survey Study

**DOI:** 10.3390/ijerph20042948

**Published:** 2023-02-08

**Authors:** Riittakerttu Kaltiala, Elias Heino, Mauri Marttunen, Sari Fröjd

**Affiliations:** 1Faculty of Medicine and Health Technology, Tampere University, 33014 Tampere, Finland; 2Department of Adolescent Psychiatry, Tampere University Hospital, 33520 Tampere, Finland; 3Vanha Vaasa Hospital, 65380 Vaasa, Finland; 4Adolescent Psychiatry, Helsinki University Hospital, University of Helsinki, 00260 Helsinki, Finland; 5Department of Health Sciences, Faculty of Social Sciences, Tampere University, 33014 Tampere, Finland

**Keywords:** transgender, adolescence, family, depression, anxiety

## Abstract

Sociodemographic and psychosocial family factors have profound implications for adolescent development, identity formation and mental health during the adolescent years. We explored the associations of sociodemographic and psychosocial family factors with transgender identity in adolescence and the role of these factors in the associations between gender identity and emotional disorders. Data from a large adolescent population survey from Finland were analysed using logistic regression models. Reporting transgender identity was associated with mother’s low level of education, accumulating family life events, lack of family cohesion, perceived lack of family economic resources and female sex. A lack of family cohesion further differentiated between adolescents reporting identifying with the opposite sex and those reporting non-binary/other gender identification. The associations between transgender identity, depression and anxiety were attenuated but did not level out when family factors were controlled for. Transgender identity in adolescence is associated with socioeconomic and psychosocial family factors that are known correlates of negative outcomes in mental health and psychosocial well-being. However, transgender identification is also associated with emotional disorders independent of these family factors.

## 1. Introduction

During adolescence, the young person progresses through various developmental tasks, the successful resolution of which is thought to contribute towards normative development and transition from childhood to adulthood [1,2]. One of these tasks is to establish emotional independence from the parents [2], while simultaneously peer relations assume a more important role in an adolescent’s life [3] as they begin to turn to their peers for social support. Despite this shift, family factors, both socioeconomic and psychosocial, decisively influence adolescent development, well-being and identity formation.

Low socioeconomic status (SES) of the family is associated with issues in peer relations [4], early and/or risky sexual behaviour [5], other risky behaviour such as substance use [6] and psychopathology [7] in the adolescent offspring. Low SES of the family is further associated with the offspring’s weaker academic performance and predicts the young person’s own lower SES achievement in adulthood [8,9]. Socioeconomic background also influences adolescent identity development [10].

Stressful events occurring in the lives of parents have a negative impact on the development, adjustment and mental health of their children [11], and recent large-scale studies have shown that the occurrence of stressful life events such as parents’ divorce, unemployment, incarceration and death is even more devastating if these occur during the offspring’s adolescent development than during the early childhood years [12]. Stressful life events during childhood and early to middle adolescence further appear to shape identity development in late adolescence and emerging adulthood [13,14,15].

Factors related to intrafamilial dynamics and parental health also impact the developing adolescent. For example, parental mental health problems, parent–child conflict, affectionless control and low family cohesion have been associated with mental health problems in adolescents [16,17,18], while appropriate parental monitoring [19], good family functioning [18] and positive family climate [20] function as protective factors against negative mental health outcomes. Parental support and family connectedness also promote identity formation well into early adulthood, while inappropriate psychological control by parents complicates it [21,22,23].

Gender identity refers to the intrinsic sense of one’s gender. In gender minorities, gender identity differs from biological sex. Individuals may identify with the opposite sex, or their perception of their gender may fall right outside the dichotomous split between male and female (non-binary gender identity). Gender identity evolves and consolidates through stages of exploration and consolidation, as do other aspects of identity [24,25]. In this article, we refer to all gender minorities as transgender. A share of transgender-identifying people also present with gender dysphoria, a marked distress related to discordance between experienced gender and biological sex [26], and may seek treatment to align bodily characteristics with their experienced gender.

In the 21st century, Western gender clinics have seen an upsurge in referrals of adolescents with features of gender dysphoria [27,28]. These adolescents commonly present with mental health problems and psychosocial difficulties in various domains of life [29], which is likely at least partly explained by gender dysphoria—a continuous distress—itself, or by the prejudice and discrimination related to atypical gender expression [24,25]. In line with an increase in treatment-seeking, population studies suggest that transgender identification may be increasing in prevalence in adolescent populations [30,31]. As a normative variation of gender identity development, transgender identity per se is not regarded as a source of distress [24,25,32]. However, comparisons of transgender-identifying adolescents in population samples and their cisgender peers indicate many disparities in health and well-being between these two populations, in domains such as suicidality [33], emotional problems [34,35,36], involvement in bullying [37] and substance use [38]. Furthermore, within the transgender group, some research suggests that psychosocial problems may be even more common among non-binary than opposite sex identifying gender minority youth [37,39].

The disparities in mental health and psychosocial well-being between transgender adolescents and their mainstream peers are often attributed to how others view, receive, accept and treat transgender individuals [40]. Research has focused particularly on difficulties in peer relationships. Much less attention has been paid to family factors, such as whether there are disparities between cisgender and transgender adolescents in socioeconomic and psychosocial family characteristics and also whether family characteristics modify the associations between gender identity and mental health problems.

Adverse parental reactions towards an adolescent’s gender non-conformity have been noted as a special risk for mental health problems in this population [41]. “Coming out” with transgender identity may disturb the parent–child relationship and family dynamics, as parents may react adversely to the adolescent’s exploration of a new gender identity or be unable to appropriately support the adolescent in coming to grips with their new gender identity [41]. Given the important role of factors related to family functioning, atmosphere and the parent–child relationship in adolescent mental health problems [17,18,19,20], disturbances in intrafamilial dynamics may expose the transgender-identifying adolescent to negative mental health outcomes. Supporting this, two US-based studies with representative samples found that parent connectedness functions as a protective factor against negative mental health outcomes among transgender youth [42,43]. Similar findings have been reported in clinical and selected samples [44,45,46,47]. On the other hand, a disproportionately large share of transgender adolescents report experiencing lack of parental support and caring [34].

The association between SES and adolescent gender identity, or the modifying effect of SES on mental health problems in gender minority youth has not been specifically addressed in research. However, a New Zealand population-based study [34] observed that a disproportionate share of transgender-identifying adolescents also belonged to the most disadvantaged SES group. Parental divorce and not living with both parents has been observed to be disproportionately common in several clinical samples of gender-referred youth in various countries [48,49], including Finland [50]. However, in one population study [51], having parents with higher education did not reduce the odds of suicidal behaviour among transgender adolescents.

As a normative variation of gender identity formation [32], transgender identity is not expected to be associated with adverse socioeconomic and psychosocial family factors. Such factors may, however, explain the association between adolescent transgender identity and mental health problems. We set out, in a large sample of Finnish adolescents, to explore whether socioeconomic and psychosocial family factors are associated with transgender identity in adolescents, and whether they modify the association between transgender identity and emotional problems, a common issue among trans youth.

In more detail, we sought answers to the following questions:Are there associations between transgender identity and socioeconomic and psychosocial family factors among adolescents aged 14–20?Are there differences in these characteristics between binary and non-binary gender minority youth?Are transgender identities associated with depression and anxiety when socioeconomic and psychosocial family factors are accounted for?

## 2. Materials and Methods

### 2.1. The School Health Promotion Study

The School Health Promotion Study (SHPS) of the National Institute for Health and Welfare is a school-based cross-sectional anonymous survey designed to examine the health, health behaviours and school experiences of teenagers. The survey questionnaire is sent to every municipality in Finland. The municipalities decide if the schools in their area will participate in the survey and the vast majority of schools do indeed participate. The survey is run primarily for health policy and administrative purposes and the data are available on request for the purposes of scientific research. The main aim of the survey is to produce national adolescent health indicators that municipalities can utilise in planning services and that can be used at a national level to assess the effectiveness of health policies. The authors obtained permission to use the data for scientific research but were not responsible for collecting it. The School Health Promotion Study has received ethical approval from the Ethics Committees of Tampere University Hospital and The National Institute for Health and Welfare.

The survey is conducted among eighth and ninth graders of comprehensive school and first and second-year students in upper secondary education which follows completion of the nine years of comprehensive school. In 2017, survey participants numbered 139,829. Of these, 48.9% (n = 68,333) reported that they were male and 50.4% (n = 70,539) that they were female. Of all respondents, 0.7% (n = 957) did not report their sex and were excluded from further analyses. We also excluded respondents aged 21 or older. Of the respondents, 53.8% were in the eighth or ninth grades of comprehensive school, 25.5% were attending upper secondary school and 20.7% vocational school. The age of respondents in the comprehensive school sample was (mean (SD)) 14.83 (0.82) years, 16.83 (0.81) years in upper secondary school and 16.74 (1.69) years in vocational school. The 2017 SHPS has been shown to be well representative of the age groups studied. The response rate among 8th and 9th graders was 63% of all enrolled students, and among upper secondary education, 51% of all enrolled students belonged to the age group [52].

### 2.2. Measures

#### 2.2.1. Sex and Gender Identity

The respondents were first asked “What is your sex?”, with response alternatives “boy” and “girl” in the comprehensive school survey, and “male”/“female” in the upper secondary education response forms. This was intended to elicit the respondent’s sex as noted in their identity documents and was the opening question of the whole survey. In this paper, we call those who indicated that they were boys/males male, and those who indicated that they were girls/females female. Later, in the section of the survey addressing health, respondents were asked about their perceived gender as follows: “Do you perceive yourself to be…”, with response options “a boy”/”a girl”/“both”/“none”/”my perception varies”. According to sex and perceived gender, the respondents were categorised into one of three gender identities: cisgender identity (indicated male sex and perceives himself as a boy, or female sex and perceives herself as a girl), opposite sex identification (male sex, perceives themself as a girl; or female sex, perceives themself as a boy), and other/non-binary gender identity (independent of sex: perceived to be both a boy and a girl, perceived to be neither a boy nor a girl, variable).

#### 2.2.2. Sociodemographic and Psychosocial Variables

Family risk factors studied as independent variables were adverse sociodemographic (low level of parental education, parental unemployment, not living with both parents) and psychosocial (accumulating family life events, lack of family cohesion, perceived economic hardship) factors.

Parental education was elicited separately for the father and mother: “What is the highest education your father/mother has completed?” The response options were “comprehensive school only”/“upper secondary school or vocational school”/“upper secondary school or vocational school and further vocational studies”/“university or university of applied sciences”. For the analyses, parental educational level was coded low if the parent had not completed more than comprehensive school.

Parental unemployment was elicited with a question “During the past year, have your parents been unemployed or laid off work?” The response alternatives were “no”/“one parent/“both parents”. For the analyses, parental unemployment was dichotomized to no (none)/yes (one or both).

Family structure was elicited with a question “Do you live in the same home with both of your parents?” with response alternatives “yes” and “no”.

Family life events were elicited by asking “Did any of the following changes occur to you during this school year?” with response alternatives “yes”/“no” regarding parental separation or divorce, parents starting a blended family, birth of a sibling, severe illness, injury or death of a family member. In the present study, we formed a sum score of life changes occurring in the family (parental separation or divorce, parents starting a blended family, birth of a sibling, severe illness, injury or death of a family member) called family life events.

The variable “lack of family cohesion” was formed as a sum score of three items: “Can you talk with your parents about matters important to you?” with response alternatives “hardly ever”/“now and then”/“fairly often”/“often”; “To what extent do you agree with the statement: our family has enough family time together” with response alternatives “completely agree”/“agree”/“neither agree nor disagree”/“disagree”/“completely disagree”; and “To what extent do you agree with: I feel that I am an important part of my family?”, with response alternatives “completely agree”/“agree”/“neither agree nor disagree”/“disagree”/“completely disagree”. These items were coded from 0–4/0–5 where bigger numbers indicate less positive perception (cannot talk, family does not have enough time together, does not feel like an important part of the family). A sum score, called lack of family cohesion, was created from the same variables.

The adolescent’s perception of their family’s economic situation was elicited with a question “How would you rate your family’s economic situation?”, with response options from 1 = very good to 5 = very poor.

#### 2.2.3. Emotional Symptoms

Depression was measured with the PHQ 2, a self-report screening tool with two questions: “During the past two weeks, how often have you bothered by” (1) “feeling down, depressed, or hopeless?” (2) “little interest or pleasure in doing things?’’ Response options for both questions were “not at all,” “for several days,” “for more than half the days,” and “nearly every day” [53]. The PHQ2 has been shown to be a reliable method for detecting depression in adolescents and young adults [54,55]. In the analyses, the sum score was dichotomised to depression (3 or more points) vs. no depression (<3). Generalised anxiety symptoms were elicited with the GAD-7, a self-report questionnaire designed to identify probable cases of generalised anxiety disorder and to assess symptom severity. The GAD-7 items describe the most prominent diagnostic features of the DSM IV generalised anxiety disorder. The GAD-7 elicits how often during the last two weeks the respondent has been troubled by each of the seven core symptoms of generalised anxiety disorder. Response options are “not at all,” “for several days,” “for more than half the days,” and “nearly every day,” scored, respectively, as 1, 2, 3 and 4. The GAD-7 has been shown to be a reliable and valid measure for detecting generalised anxiety disorder in primary care and in the general population [56]. In the analyses, the sum score of these seven items was first used as a continuous variable and then dichotomised to moderate or severe anxiety (10+ points) vs. no anxiety (<10) [57,58].

#### 2.2.4. Implausible, Likely Facetious Responding

It has been demonstrated that some adolescents deliberately mispresent themselves in survey studies, exaggerating their belonging to minorities as well as their problem behaviours, symptoms and psychosocial problems [59]. Consequently, the proportion of those reporting belonging to minorities (such as disabled adolescents, immigrants, sexual minorities) appears implausibly high and associations between minority status and psychosocial problems appear excessive. In relation to gender identity, such overestimation may risk a perception in society that gender variant youth are victims rather than active subjects participating in building the contemporary adolescent community. Particularly considering the excessive media coverage of gender identity issues [60], gender identity is likely to be a topic which tempts adolescents to give facetious responses.

Excluding respondents reporting unlikely combinations of extreme responses has been shown to be an appropriate method for controlling for such facetious responding [30,59]. In line with this, respondents reporting implausibly young age for being enrolled in the grades studied (<13 years), implausible shortness or height (extreme outliers) or those who were calculated to have extreme BMI (<10 or >40) or reporting both extremely poor hearing, sight and mobility were classified as facetious responders (for a detailed description, see Kaltiala-Heino and Lindberg, 2019). Being classified as a facetious respondent was strongly associated with reporting transgender identity in these data [30]. These respondents (3.5%) were excluded from further analyses. The final sample numbered 130,322 (63,764 males and 66,558 females), mean (sd) age 15.8 (1.3) years. There was occasional non-response to study variables. The sample description is given in Table 1.

### 2.3. Statistical Analyses

Associations between family variables and reporting transgender identity were studied using logistic regression. Because the associations between family characteristics and adolescent gender identity are a largely unexplored field, the analyses are exploratory and are not tailored to test pre-defined theoretical models. Gender identity (cisgender vs. transgender) was entered as the dependent variable. First, age, sex and sociodemographic family variables were entered as independent variables. Next, psychosocial family variables were added. A similar procedure was applied to transgender-identifying adolescents only, entering gender identity (opposite sex identification vs. non-binary/other) as the dependent variable. Odds Ratios (OR) with 95% confidence intervals (CI) are reported.

In the next step, first depression (yes/no) and then anxiety (yes/no) were entered each in turn as the dependent variables. As independent variables, first gender identity was entered (cisgender vs. opposite sex identification vs. nonbinary/other gender identity) controlling for age, sex and the other emotional disorder (anxiety when analysing associations with depression and vice versa) (Step 1); then, family sociodemographic variables were added (Step 2), and finally further family psychosocial variables (Step 3). Odds Ratios (OR) with 95% confidence intervals (CI) are reported.

Due to the large data size, the cut-off point for statistical significance was set at *p* < 0.001.

## 3. Results

Of the respondents, 96.2% reported cisgender identity, 0.6% opposite sex identification and 3.2% non-binary/other gender identity. Transgender identity was more commonly reported by female respondents (0.8% opposite sex, 3.6% non-binary/other vs. 0.4% opposite sex and 2.8% non-binary/other, *p* < 0.001).

Table 2 presents the associations between family variables and group membership (transgender vs. cisgender). Most of the associations first seen between SES variables and transgender identity levelled out when psychosocial variables were added. When all study variables were entered simultaneously, the following family variables were statistically significantly associated with transgender identity: mother having only basic education (OR (95% CI) = 1.3 [1.1–1.4], *p* < 0.001), accumulating family life events (OR (95% CI) = 1.3 [1.2–1.3], *p* < 0.001), lack of family cohesion (OR (95% CI) = 1.2 [1.1–1.1.2], *p* <0.001).

Within the transgender-identifying group, a perceived lack of family cohesion was associated with non-binary/other gender identity (OR (95% CI) = 1.1 (1.1–1.2), *p* < 0.001), but otherwise the family variables studied did not differentiate between gender identity subgroups.

Table 3 presents the associations between gender identity and depression, controlling for age, sex and anxiety (Step 1), adding sociodemographic family variables (Step 2) and further adding psychosocial family variables (Step 3). The first seen associations between transgender identities and depression grew weaker, particularly when psychosocial family variables were added into the model. In the final analysis, after accounting for all factors, opposite sex identification was borderline significantly, and non-binary/other gender identity were statistically significantly associated with depression. In addition, female sex, parental unemployment, all the psychosocial family factors studied, and anxiety were statistically significantly associated with depression.

Table 4 presents the associations between gender identity and anxiety, controlling for depression and the remaining variables as in the analyses focusing on depression. The associations between opposite-sex identification and anxiety attenuated below the set cut-off for statistical significance when psychosocial family variables were added, but the associations between non-binary/other gender identity attenuated after adding socio-demographics and increased again after adding psychosocial family factors. In the final analysis, anxiety was statistically significantly associated with non-binary/other gender identity, age, female sex and depression and also with all the psychosocial family variables studied.

## 4. Discussion

Contrary to what was hypothesised, adolescent transgender identity was, in this uniquely large population study, associated with mother’s low education, accumulating family life events, lack of family cohesion and perceived lack of family economic resources, as well as with female sex. Lack of family cohesion further differentiated between adolescents reporting opposite-sex identification and non-binary/other gender identity. Earlier studies have, to the best of our knowledge, not specifically focused on sociodemographic and psychosocial family characteristics of transgender-identifying youth in the population. We further explored the role of sociodemographic and psychosocial family factors for the association between transgender identities and emotional symptoms. Partially, as expected, associations between transgender identities and emotional problems grew weaker when socioeconomic and psychosocial family factors were accounted for. The associations were, nevertheless, not completely levelled out by accounting for family factors, particularly not those between non-binary identity and depression/anxiety.

Earlier research on transgender identity in adolescent population studies or clinical samples of adolescents with gender dysphoria have rarely focused specifically on family correlates of transgender identity in youth, although they have occasionally reported low family SES [34] and disproportionately common parental divorce [48,50]. Kozlowska et al. [49,61], however, reported that the backgrounds of children and adolescents presenting with clinical gender dysphoria were much more commonly characterised by not living with both parents, adverse life events, family conflict and high-risk attachment patterns than those of age and sex matched controls. They concluded that developmental pathways to adolescent transgender identity and gender dysphoria are at least partly shaped by stressful life events, lack of family stability and cohesion and socioeconomic disadvantage [61]. The present study among adolescents in the general population similarly found associations between adverse sociodemographic and psychosocial family factors and adolescent transgender identities. Being cross-sectional, our study cannot shed light on causal pathways behind the associations. Family cohesion, especially, could grow weaker as a result of the family’s reaction to the adolescent’s disclosure of transgender identity, but lack of family cohesion can also be a factor that has an impact on the adolescent’s identity development.

Greater perceived lack of family cohesion was the only family variable that distinguished between transgender identities, opposite-sex identification and non-binary/other gender identity. The non-binary/other group reported a greater lack of family cohesion. Parental reactions towards non-binary identity could be more ambivalent and negative than towards opposite-sex identification, and this could estrange the adolescent from their parents even more than disclosure of a binary trans identity. However, as the present study is cross-sectional, causal assumptions must be made with great caution.

The second aim of the present study was to explore the role of family characteristics on the association between transgender identities and adolescent depression and anxiety. When socioeconomic and psychosocial family variables were accounted for, the associations between transgender identity and emotional variables attenuated but were not completely levelled out: opposite sex identification persisted as borderline significantly associated with depression and non-binary/other gender identity, and as statistically significantly associated with both depression and anxiety. Unfavourable family characteristics explain the associations between adolescent gender identity and emotional problems in adolescence to considerable extent. Earlier studies in population [42,43] and clinical and selected samples ([44,45,46,47] have suggested that family support, connectedness and cohesion are protective of mental health problems among trans youth, with a stronger effect than peer relationships. Our results showed that when the role of family characteristics is controlled for, mental health disparities between cisgender and opposite sex identifying adolescents almost disappear.

Some earlier studies [37,39], although not all [51], have suggested that among trans youth, non-binary individuals struggle with even more emotional and relational issues than their opposite-sex identifying peers. This was the case in the present study. Identity formation entails exploration of different identities followed eventually by identity consolidation [62,63]. Exploration takes place first in breadth, across identity options, and later in depth to alternatives that start to emerge as optimal [64]. These are normative developmental processes. However, when exploration in breadth fails to progress to exploration in depth and further into identity consolidation, psychopathology may emerge [64]. Non-binary individuals identify with neither female nor male sex, or their perception of their gender identity may fluctuate over time. This may suggest that they spend more time in the process of exploration in breadth, possibly exposing them to the development of emotional issues. Opposite-sex identifying adolescents, on the other hand, may have more often proceeded from exploration to commitment, which promotes well-being. This could be one factor contributing to mental health disparities between non-binary and opposite-sex identifying youth.

### Methodological Considerations

A strength of the present study is the uniquely large, nationally representative non-selected adolescent sample comprising participants from early to late adolescence. In Finland, comprehensive school was in 2017 compulsory from age 7 to 16 (since 2021 until 18). Over 99% of children were enrolled, and over 95% of the adolescents proceeded to vocational or upper secondary schools [65]. This further ensures that our sample is representative. Variation in participation rates between regions is a result of the municipalities’ decisions on whether or not to participate in the SHPS, and as such is not likely to bias findings on the phenomena reported on in this paper. A limitation is that the study did not reach adolescents who were not at school on the survey day. Such absences are known to account for 10–15% of those enrolled in schools. Another limitation is that our data did not include adolescents not in mainstream education. According to theories of identity development [62,63], those outside education could be assumed to display identity diffusion more commonly than their peers, who are integrated into age-appropriate commitments, but we are not aware of studies focusing on gender identity among them.

As has been recommended [35,36], we used a two-step approach to identify subjects with different gender identities, eliciting sex and gender perception separately. Eliciting sex was the opening question of the whole survey, while perceived gender was elicited in the section on health after eliciting perceived health, height and weight. The possibility that some respondents identifying strongly with the opposite sex already reported perceived gender rather than natal sex in the first step cannot be controlled for, as has been the case in earlier studies of this kind. This is a limitation inherent in the anonymous survey method and a limitation in the present study. Most likely, the respondents understood that sex as noted in identity documents was elicited at the beginning of the survey, as this is common practice in any official documents and forms in Finland. A further limitation of this approach is that identity processes were not explored. Simply taking stated identification as identity entails a risk of overestimation as achieved identity, when it may rather represent stages of identity exploration.

Depression and anxiety were elicited with validated methods, and family variables had all been used in earlier Finnish studies on adolescent well-being and mental health, which adds to the reliability of the study.

Excluding implausible and potentially facetious responding is a further strength of the present study. Gender identity has earlier been shown to be a topic susceptible to facetious responding in these data [30]. Using a technique to exclude facetious responding, as proposed by Robinson-Cimpian [59], is a strength of the present study.

Our study also has several weaknesses. As this is a cross-sectional study, no causality can be determined. The experience of lack of family cohesion could emerge after the onset of transgender identity. For example, parents may react adversely [41,66] or adolescents may not feel comfortable about disclosing their gender identity to their parents. On the other hand, a lack of family cohesion may expose adolescents to emotional issues [18]. Perceived financial difficulties in the family could also reflect true adversity, and as such, even a predisposing factor for emotional difficulties could complicate identity development, but also negative attribution bias related to distress due to identity struggles. However, parental education is unlikely to be affected by the adolescent offspring’s identity, and the elicited life events were also relatively little open to interpretation.

Further, we did not elicit whether respondents had disclosed their gender identity or lived in a role that reflects their gender identity, or whether their parents knew about their transgender identity. Concealment of one’s gender identity could subject individuals to emotional problems [40] and the literature suggests that disclosing one’s gender identity has a positive impact on mental health [67]. However, disclosure may also risk adverse reactions from the environment, including parents. Therefore, we were unable to compare those transgender adolescents who have disclosed their gender identity with those who have not, and to explore how the disclosure of one’s gender identity mediates associations between transgender identity, emotional problems and experiences of a lack of family cohesion; thus, our data are limited. Finally, we were not able to explore to what extent the adolescents had actually explored identity options and made commitments, and to what extent the stated transgender identity therefore represented achieved, consolidated identity [62,63,64].

## 5. Conclusions

Transgender identity in adolescence is associated with socioeconomic and psychosocial family factors that are known correlates of negative outcomes in mental health and psychosocial well-being. A part of transgender identification in adolescence may represent developmental challenges. On the other hand, some unfavourable family characteristics such as lack of family cohesion may develop as an adverse reaction to the adolescent offspring’s transgender identity. Family adversities must in any case be considered by those healthcare workers, teachers and others working with transgender-identifying adolescents, and appropriate interventions need to be offered.

Transgender identities, particularly non-binary, have independent associations with depression and anxiety even when family factors are accounted for. The biological, relational and cultural underpinnings and consequences of the development of transgender identity in adolescence in the contemporary society warrant further research. Future research also needs to explore the associations between socioeconomic and psychosocial family characteristics and adolescent gender identity development in longitudinal designs.

## Figures and Tables

**Table 1 ijerph-20-02948-t001:** Sample description.

	% (n/N)	Mean (sd); Median; Range
Sex		
Male	48.9 (63,764/1,303,222)	
Female	51.1 (66,558/130,322)	
Age		15.8 (1.3); 16; 13–20
Gender identity		-
Cisgender	95.3 (124,219/129,126)	
	0.6	
Opposite-sex identification	(813/129,126)	
	3.1	
Non-binary/other	(4094/130,322)	
Not living with both parents	31.1 (40,553/125,870)	-
Mother’s education low	5.2 (6722/119,498)	-
Father’s education low	8.2 (10,657/117,314)	-
Parental unemployment	29.9 (38,923/124,737)	-
Depression	13.0 (16,949/128,879)	1.0 (1.5); 0; 0–6
Anxiety	11.3 (14,752/129,076)	3.8 (4.6); 2; 0–21
Family life events	-	0.3 (0.6); 0; 0–4
Lack of family cohesion	-	2.5 (2.0); 2; 0–10
Perceived economic resources	-	2.2 (0.9); 2; 1–5

**Table 2 ijerph-20-02948-t002:** Associations between socioeconomic and psychosocial family variables and transgender identity among 14–20-year-old Finnish adolescents. In Step 1, sociodemographic family variables are studied while controlling for age and sex. In Step 2, psychosocial family variables are added. Odds Ratios (95% CI) are reported.

	Step 1		Step 2	
	OR (95% CI)	*p*	OR (95% CI)	*p*
Age	1.0 (1.0–1.0)	0.07	1.0 (1.0–0.1)	0.02
Sex				
Male	ref.		ref.	
Female	**1.5 (1.4–1.6)**	**<0.001**	**1.3 (1.2–1.4)**	**<0.001**
Socioeconomic family variables				
Not living with both parents	**1.2 (1.2–1.3)**	**<0.001**	1.0 (1.0–1.1)	0.3
Mother’s education low	**1.4 (1.3–1.6)**	**<0.001**	**1.3 (1.13–1.43)**	**<0.001**
Father’s education low	1.2 (1.1–1.3)	0.002	1.1 (1.0–1.3)	0.04
Parents’ unemployment	**1.3 (1.2–1.4)**	**<0.001**	1.1 (1.1–1.2)	0.001
Psychosocial family variables				
Family life events			**1.3 (1.2–1.4)**	**<0.001**
Lack of family cohesion			**1.2 (1.2–1.3)**	**<0.001**
Perceived economic resources			**1.1 (1.1–1.1)**	**<0.001**

Note: bold typeface for statistically significant values.

**Table 3 ijerph-20-02948-t003:** Associations between gender identity and depression.

	Step 1		Step 2		Step 3	
	OR (95% CI)	*p*	OR (95% CI)	*p*	OR (95% CI)	*p*
Age	1.0 (1.0–1.0)	0.4	1.00 (1.0–1.0)	0.7	0.1 (1.0–1.0)	0.5
Sex						
Male	ref	ref	ref	ref		
Female	**1.7 (1.7–1.8)**	**<0.001**	**1.7 (1.6–1.8)**	**<0.001**	**1.5 (1.4–1.6)**	**<0.001**
Gender identity						
Cisgender	ref	ref	ref	ref		
Opposite-sex identification	**1.8 (1.4–2.2)**	**<0.001**	**1.7 (1.3–2.1)**	**<0.001**	1.5 (1.2–1.9)	0.001
Non-binary/other	**3.1 (2.8–3.4)**	**<0.001**	**3.0 (2.7–3.3)**	**<0.001**	**2.3 (2.1–2.5)**	**<0.001**
Anxiety	**29.2 (27.9–30.5)**	**<0.001**	**27.8 (26.5–29.1)**	**<0.001**	**21.7 (20.7–22.8)**	**<0.001**
Socioeconomic family variables	-					
Not living with both parents			**1.3 (1.2–1.3)**	**<0.001**	1.1 (1.0–1.1)	0.01
Mother’s education low			1.2 (1.1–1.3)	0.002	1.0 (0.9–1.2)	0.4
Father’s education low			1.1 (1.0–1.2)	0.003	1.0 (1.0–1.1)	0.3
Parental unemployment			**1.4 (1.3–1.4)**	**<0.001**	**1.1 (1.1–1.2)**	**<0.001**
Psychosocial family variables	-		-			
Family life events					**1.1 (1.1–1.2)**	**<0.001**
Lack of family cohesion					**1.4 (1.4–1.4)**	**<0.001**
Perceived economic resources					**1.2 (1.2–1.2)**	**<0.001**

Note: bold typeface for statistically significant values.

**Table 4 ijerph-20-02948-t004:** Associations between gender identity and anxiety.

	Step 1		Step 2		Step 3	
	OR (95% CI)	*p*	OR (95% CI)	*p*	OR (95% CI)	*p*
Age	**1.1 (1.1–1.1)**	**<0.001**	**1.1 (1.1–1.1)**	**<0.001**	**1.1 (1.1–1.1)**	**<0.001**
Sex (natal)						
Male	ref	ref	ref	ref		
Female	**3.5 (3.3–3.7)**	**<0.001**	**3.6 (3.4–3.7)**	**<0.001**	**3.3 (3.2–3.5)**	**<0.001**
Gender identity						
Cisgender	ref	ref	ref	ref		
Opposite-sex identification	**1.5 (1.2–1.9)**	**<0.001**	**1.5 (1.2–1.9)**	**<0.001**	1.4 (1.1–1.8)	0.01
Non-binary/other	**1.6 (1.4–1.7)**	**<0.001**	**1.2 (1.1–1.3)**	**<0.001**	**1.4 (1.3–1.6)**	**<0.001**
Depression	**29.2 (28.0–30.5)**	**<0.001**	**27.9 (26.6–29.2)**	**<0.001**	**21.7 (20.7–22.8)**	**<0.001**
Socioeconomic family variables						
Not living with both parents	-		**1.2 (1.1–1.3)**	**<0.001**	1.1 (1.0–1.1)	0.03
Mother’s education low	-		1.0 (0.9–1.1)	0.6	1.0 (0.9–1.1)	0.4
Father’s education low	-		1.1 (1.0–1.2)	0.03	1.1 (1.0–1.1)	0.3
Parents’ unemployment	-		**1.2 (1.1–1.3)**	**<0.001**	1.1 (1.0–1.1)	0.01
Psychosocial family variables						
Family life events	-		-		**1.2 (1.2–1.3)**	**<0.001**
Lack of family cohesion	-		-		**1.2 (1.1–1.2)**	**<0.001**
Perceived economic resources	-		-		**1.1 (1.1–1.2)**	**<0.001**

Note: bold typeface for statistically significant values.

## Data Availability

Researchers can request data from Findata https://findata.fi/.

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
