# Peer review of "Family Characteristics, Transgender Identity and Emotional Symptoms in Adolescence: A Population Survey Study"

_ijerph, 2023, doi:10.3390/ijerph20042948_

Round 1

Reviewer 1 Report

this is an important and carefully conducted study, and thoughtfully presented. My one problem is that the description of the sample of students does not include a response rate...the paper gives the number of students who participated but not the number who refused to complete the questionnaire, or indeed the proportion of schools which refused to participate. The authors claim the study is representative  (I think assuming  it must be so because of the large number of participants) but it is not necessarily so unless we know more about the school refusals and the student refusals. 

Author Response

Reviewer 1

this is an important and carefully conducted study, and thoughtfully presented. My one problem is that the description of the sample of students does not include a response rate...the paper gives the number of students who participated but not the number who refused to complete the questionnaire, or indeed the proportion of schools which refused to participate. The authors claim the study is representative  (I think assuming  it must be so because of the large number of participants) but it is not necessarily so unless we know more about the school refusals and the student refusals. 

RE: We added response rates for pupils in 8th and 9th grade (63% of all enrolled in the country), and for students in upper secondary education (51% of all enrolled in the country). There was some variation between regions in what proportion of students were included, but not remarkable. Unfortunately, it is not known how many personally refused to participate in the survey, because SHPS is run completely anonymously. It is known from earlier years that 10-15% of students are absent on any given day. We now also comment on this source of missing our in methodological considerations.

Reviewer 2 Report

The sample used is fairly representative of the population under study.

The control and elimination of anomalous or suspected false answers is clearly stated and, in principle, justified.

The contribution that this study can make to the development of measures to support and facilitate the lives of the adolescent population justifies its implementation.

I believe it is a well-conducted research.

Among the suggestions for improvement of this work I would like to point out:

- Show a table with the descriptive indices of the sample, even if they are reported in the text.

- Show the tables of the logistic regression analysis more clearly. Perhaps using a table for each proposed model and another to compare them. The table presented is a little difficult to read.

- Justify more explicitly which of the models is more appropriate according to methodological and theoretical criteria.

- Although at all times the article talks about the association between the variables and correctly avoids talking about causal relationships, in order to avoid misinterpretations by the reader, the conclusions should include the idea that transgender identity or non-binary gender identity is not necessarily a "consequence" of the variables studied, since factors such as lack of understanding, non-acceptance or social and family rejection can unfortunately derive from not knowing how to accept the adolescent's gender option. In other words, the relationship may be bidirectional. 

Non-acceptance or rejection by their peers or their own family seems logical to produce depressive symptoms or generalised anxiety, not necessarily the fact of not being cisgender.

I think that the introduction at the beginning of the article explains it well, but perhaps some of the clarifications suggested in the Conclusions should be made.

Author Response

Reviewer 2

The sample used is fairly representative of the population under study.

The control and elimination of anomalous or suspected false answers is clearly stated and, in principle, justified.

The contribution that this study can make to the development of measures to support and facilitate the lives of the adolescent population justifies its implementation.

I believe it is a well-conducted research.

Among the suggestions for improvement of this work I would like to point out:

- Show a table with the descriptive indices of the sample, even if they are reported in the text.

RE: Such table is now added. We removed corresponding comments from the text in order to not repeat same information in text and tables.

-Show the tables of the logistic regression analysis more clearly. Perhaps using a table for each proposed model and another to compare them. The table presented is a little difficult to read.

and

- Justify more explicitly which of the models is more appropriate according to methodological and theoretical criteria.

RE: We apologize for inconvenience! We had presented tables with landscape layout and they had indeed taken a slightly difficult-to-read form in vertical pages. We have now edited the tables so that they are more pleasant to read. We also added a comment in statistical analysis that we are not testing the fit of theoretically motivated models but running exploratory analyses, given the lack of earlier research on the topic of family characteristics and adolescent gender identity. In order to avoid confusing by referring to models – as if there were theoretical models to test – we changed the steps of analysis to “steps” and made minor editions to frazing in the text.

- Although at all times the article talks about the association between the variables and correctly avoids talking about causal relationships, in order to avoid misinterpretations by the reader, the conclusions should include the idea that transgender identity or non-binary gender identity is not necessarily a "consequence" of the variables studied, since factors such as lack of understanding, non-acceptance or social and family rejection can unfortunately derive from not knowing how to accept the adolescent's gender option. In other words, the relationship may be bidirectional. 

RE: We checked carefully that we had considered all detected associations both as risk factors for and as consequences of adolescent identity development, and emphasized in all entries that our data, being cross-sectional, cannot support conclusions on causality. We noticed that this could be better emphasized in Conclusion and edited it as follows:

“Transgender identity in adolescence is associated with socioeconomic and psychosocial family factors that are known correlates of negative outcomes in mental health and psychosocial well-being. A part of transgender identification in adolescence may represent developmental challenges. On the other hand, some unfavorable family characteristics such as lack of family cohesion may develop as adverse reaction to adolescent offspring’s transgender identity. Family adversities must in any case be considered by those healthcare workers, teachers and others working with transgender identifying adolescents, and appropriate interventions need to be offered.

Transgender identities, particularly non-binary, have independent associations with depression and anxiety even when family factors are accounted for. The biological, relational and cultural underpinnings and consequences of the development of transgender identity in adolescence in the contemporary society warrant further research. Future research also needs to explore the associations between socioeconomic and psychosocial family characteristics and adolescent gender identity development in longitudinal designs.”

Otherwise, in Discussion, we mention on lines 337-341:

“…s. Being cross-sectional, our study cannot shed light on causal pathways behind the associations. Family cohesion especially could grow weaker as family’s reaction to adolescent’s disclosure of transgender identity, but lack of family cohesion can also be a factor that has an impact on adolescent’s identity development. “

and on lines 344-347:

“Parental reactions towards non-binary identity could be more ambivalent and negative than towards opposite-sex identification, and this could estrange the adolescent from their parents even more than disclosure of a binary trans identity. However, as the present study is cross-sectional, causal assumptions must be made with great caution.”

Non-acceptance or rejection by their peers or their own family seems logical to produce depressive symptoms or generalised anxiety, not necessarily the fact of not being cisgender.

RE: Yes, exactly,and this is why we assumed that when family variables are controlled for, differences in depression and anxiety between trans – and cisgender adolescents might no more emerge. But the differences were not completely levelled out by accounting for family variables. However, we now emphasize more the role of family variables for depression/ anxiety on lines 355-357: “Unfavourable family characteristics explain the associations between adolescent gender identity and emotional problems in adolescence to considerable extent. Earlier studies…”

I think that the introduction at the beginning of the article explains it well, but perhaps some of the clarifications suggested in the Conclusions should be made.                                                               

RE: Conclusion was edited.